# Intron Regions as Genetic Markers for Population Genetic Investigations of *Opisthorchis viverrini* sensu lato and *Clonorchis sinensis*

**DOI:** 10.3390/ani13203200

**Published:** 2023-10-13

**Authors:** Chairat Tantrawatpan, Wanchai Maleewong, Tongjit Thanchomnang, Warayutt Pilap, Takeshi Agatsuma, Ross H. Andrews, Paiboon Sithithaworn, Weerachai Saijuntha

**Affiliations:** 1Division of Cell Biology, Department of Preclinical Sciences, Faculty of Medicine, and Center of Excellence in Stem Cell Research, Thammasat University, Rangsit Campus, Khlong Nueng 12120, Thailand; talent3003@yahoo.com; 2Department of Parasitology, Faculty of Medicine, Khon Kaen University, Khon Kaen 40002, Thailand; wanch_ma@kku.ac.th (W.M.); paibsit@gmail.com (P.S.); 3Mekong Health Science Research Institute, Khon Kaen University, Khon Kaen 40002, Thailand; 4Faculty of Medicine, Mahasarakham University, Kham Riang 44000, Thailand; tongjit.t@msu.ac.th; 5Walai Rukhavej Botanical Research Institute, Mahasarakham University, Kham Riang 44150, Thailand; warayuttpilap@gmail.com; 6Center of Excellence in Biodiversity Research, Mahasarakham University, Kham Riang 44150, Thailand; 7Department of Environmental Medicine, Kochi Medical School, Kochi University, Oko, Nankoku 783-8505, Kochi, Japan; agatsuma@kochi-u.ac.jp; 8Department of Surgery & Cancer, Faculty of Medicine, Imperial College, South Kensington Campus, London SW7 2AZ, UK; rhandrews@gmail.com; 9Cholangiocarcinoma Research Institute, Faculty of Medicine, Khon Kaen University, Khon Kaen 40002, Thailand

**Keywords:** zoonoses, cat, dog, gastrointestinal parasite, intron, genetic variation, liver fluke

## Abstract

**Simple Summary:**

The zoonotic liver flukes *Opisthorchis viverrini* and *Clonorchis sinensis* infect small mammals, such as cats, dogs, pigs, rodents, and rabbits, as well as humans. Human infection subsequently develops into bile duct malignancy, also referred to as cholangiocarcinoma (CCA). Understanding the molecular systematics and population genetics of these liver flukes has an important role in prevention and control, and is important in comprehending their roles in zoonotic transmission. Different molecular markers have varying evolution rates and contain different genetic information. Polymorphic genetic markers are necessary and more suitable for such investigations. Therefore, we screened seven intron regions of the taurocyamine kinase gene (TK) to determine their potential as genetic markers for population genetic investigations of the liver flukes *O. viverrini* and *C. sinensis* which were collected from a range of geographical isolates and animal hosts. We identified a suitable intron region of TK, i.e., intron 5 of domain 1 (TkD1Int5) as having the most potential as a polymorphic marker. Results showed that TkD1Int5 is effective in examining the genetic variation and heterozygosity of *O. viverrini* and *C. sinensis*, but further studies are required to better understand the role of different species of animals as reservoir hosts of these zoonotic liver flukes.

**Abstract:**

Opisthorchiasis and clonorchiasis are prevalent in Southeast and Far-East Asia, which are caused by the group 1 carcinogenic liver flukes *Opisthorchis viverrini* sensu lato and *Clonorchis sinensis* infection. There have been comprehensive investigations of systematics and genetic variation of these liver flukes. Previous studies have shown that *O. viverrini* is a species complex, called “*O. viverrini* sensu lato”. More comprehensive investigations of molecular systematics and population genetics of each of the species that make up the species complex are required. Thus, other polymorphic genetic markers need to be developed. Therefore, this study aimed to characterize the intron regions of taurocyamine kinase gene (TK) to examine the genetic variation and population genetics of *O. viverrini* and *C. sinensis* collected from different geographical isolates and from a range of animal hosts. We screened seven intron regions embedded in TK. Of these, we selected an intron 5 of domain 1 (TkD1Int5) region to investigate the genetic variation and population genetics of theses liver flukes. The high nucleotide and haplotype diversity of TkD1Int5 was detected in *O. viverrine*. Heterozygosity with several insertion/deletion (indel) regions were detected in TkD1Int5 of the *O. viverrine* samples, whereas only an indel nucleotide was detected in one *C. sinensis* sample. Several *O. viverrine* samples contained three different haplotypes within a particular heterozygous sample. There were no genetic differences between *C. sinensis* isolated from various animal host. Heterozygous patterns specifically detected in humans was observed in *C. sinensis*. Thus, TkD1Int5 is a high polymorphic genetic marker, which could be an alternative marker for further population genetic investigations of these carcinogenic liver flukes and other related species from a wide geographical distribution and variety of animal hosts.

## 1. Introduction

Liver flukes of the family Opisthorchiidae are among the medically important foodborne trematodes. Currently, three principal species are recognized as pathogens causing human diseases: *Opisthorchis viverrine*, *Opisthorchis felineus*, and *Clonorchis sinensis* [1]. Up to 680 million people around the world are at risk of infection [2]. Of these, two species, namely *O. viverrini* and *C. sinensis*, have distributions in Asia. For instance, *O. viverrini* is currently found in Thailand, Cambodia, Lao People’s Democratic Republic (PDR), Myanmar, and Vietnam, with approximately 10 million people having opisthorchiasis, whereas *C. sinensis* is found in parts of Russia, China, Korea, and Vietnam, with an estimated 35 million clonorchiasis cases [1]. Both are classified as group I carcinogens because they are the causative agents of bile duct cancer (cholangiocarcinoma; CCA) [1,3]. At present, about 20,000 people die of CCA every year in the northeast of Thailand alone [4].

The life cycles of all two species are very similar, beginning snails as the first intermediate host with a usually low prevalence of infection, fish as second intermediate hosts with substantially higher levels of infection, and usually a carnivorous mammal as final hosts [4]. Most recently, there is evidence of genetic similarity between liver flukes recovered from mammalian hosts, especially cats and *O. viverrini* in people [5,6]. Therefore, the complete elimination of infection may not be possible if domestic cat and dogs act as reservoir hosts, thus maintaining the source of flukes, therefore playing a critical role in maintaining their life cycle [4]. Thus, it is recommended that transdisciplinary research, including systematics and population genetics of the flukes and their life cycle hosts, should be undertaken to combat the liver flukes, therefore contributing to the reduction in CCA, particularly in endemic areas [4].

The comprehensive molecular systematics and population genetics investigations of *O. viverrini* have previously been undertaken, and cryptic species were discovered in 2007 [7]. Investigations of genetic variation of *C. sinensis* have also been reported from Russia, Vietnam, China, and Korea [6]. Some genetic markers used previously were unsuitable and unreliable for systematics and population genetic studies, such as markers in mitochondrial DNA genes, and some conserved regions in nuclear genes, due to observed low variation reported in some studies [7]. However, various molecular markers show different evolutionary rates and contain different genetic information. Compared to multi-copy sequences, single-copy genes may be more advantageous in identifying heterozygotes [8]. Thus, other polymorphic molecular markers, such as the sequence of intron regions of taurocyamine kinase gene (TK) need to be characterized and used to explore the genetic variation and population genetics of these liver flukes [7].

Phosphagen kinases (PKs) are phosphotransferases essential for cellular energy metabolism. In cells with high and fluctuating energy turnover rates, these highly conserved enzymes catalyze the reversible phosphate transfer between ATP and guanidine molecules [9]. Eight PKs are currently known, with creatine kinase (CK) as the sole PK in vertebrates. Arginine kinase (AK), glycocyamine kinase (GK), hypotaurocyamine kinase (HTK), lumbricine kinase (LK), opheline kinase (OK), thalessemine kinase (ThK), and taurocyamine kinase (TK) are PKs that are present in a wide range of invertebrate taxa [10]. The phosphagen kinase gene in trematodes refers to a specific gene, referred to as the taurocyamine kinase gene, which encodes an enzyme that plays a key role in the energy metabolism of trematode parasites [11]. Studying TK in trematodes involves characterizing its sequence, structure, expression patterns, and functional properties for identifying potential vulnerabilities that can be targeted for therapeutic interventions [11]. Not only functional genes (exon regions) but also intron regions have been successfully used as a genetic marker(s) to investigate the systematics and genetic variation of various organisms [12,13]. 

Non-coding introns are now commonly used in molecular phylogenetics and population genetics in an attempt to construct phylogenetic trees, and determine heterozygosity, DNA recombination, and genetic hybridization [14,15,16,17]. Compared to coding regions, introns predict the acquisition of a large number of independent parsimony-informative characters from most sites equally, associated with less homoplasy and lower transition: transversion ratios. Moreover, it must be acknowledged that diploid spliceosomal intron alleles have an average effective population size four times that of mitochondrial DNA (mtDNA) and in animals, introns mutate at approximately one quarter the rate of animal mtDNA [18,19].

Based on several previous publications, the intron regions of TK could offer an adequate resolution to examine genetic variation, genetic differentiation, genetic relationships, and heterozygosity of medically important parasitic trematodes [15,16,17]. For example, the bridge intron (TkBridgeInt) and intron 4 of domain 2 (TkD2Int4) regions have been applied to differentiate *Fasciola gigantica* and *F. hepatica*, as well as to explore genetic recombination in a *Fasciola* species (intermediate form), and in the case of using TkBridgeInt and TkD2Int4 regions, providing evidence of hybridization between *F. gigantica* and *F. hepatica* [15]. The second intron of domain 1 (TkD1Int2) was successfully used to differentiate species of the lung fluke genus *Paragonimus* [16], and evidence of the DNA recombination of *P. heterotremus* and *P. pseudoheterotremus* suggested incomplete lineage sorting [16]. Moreover, high polymorphism and a heterozygous TkD1Int5 region were detected in the intestinal flukes *Echinostoma* spp., which could potentially be used to differentiate *E. miyagawai* and *E. revolutum* [17]. All previous reports have shown that the intron region of TK has a potential role as a polymorphic genetic marker to explore the systematics and population genetics, especially heterozygosity, DNA recombination, and genetic hybridization, in these medically important trematodes [15,16,17]. Therefore, this study aimed to characterize and use the TK’s intron region(s) as a molecular marker(s) for the population genetic investigations of *O. viverrini* sensu lato and *C. sinensis* isolated from various geographical isolates and animal hosts.

## 2. Materials and Methods

### 2.1. DNA Samples

The DNA of *O. viverrini* from Thailand and Lao PDR were provided by Prof. Paiboon Sithithaworn. The DNA of *C. sinensis* from Vietnam included samples from *Haplorchis taichui*, *Haplorchis pumilio*, and *Stellentchasmas fulcatus* which were provided by Prof. Wanchai Maleewong. The DNA of *C. sinensis* was collected from different hosts in China and was provided by Prof. Takeshi Agatsuma. A total of 147 *O. viverrini* and 45 *C. sinensis* DNA samples from different geographical localities and animal hosts in Thailand, Lao PDR, Vietnam, and China were used for comparative analyses in this study. The DNA samples of *H. pumilio*, *H. taichui*, and *S. fulcatus* were used to test cross-reactivity for TkD1Int5 amplification.

### 2.2. Primer Design

We designed the primer pairs to anneal the conserved flanking regions (exon) based on the full-length TK sequence of *C. sinensis* accession number JX435779 [11] for amplification by polymerase chain reaction (PCR) of seven intron regions. The resultant primers used to amplify three intron regions of domain 1 were intron 1 (TkD1Int1), intron 2 (TkD1intron2), and intron 5 (TkD1Int5), and three regions of domain 2, namely intron 2 (TkD2Int2), intron 3 (TkD2Int3), intron 4 (TkD2Int4), including the bridge intron (TkBridInt) (Figure 1). We also design a second primer pair (F2 and R2) to amplify some intron regions by a second PCR in cases where we were unable to amplify or there were low concentrations of PCR product amplified by the first primer pair (F1 and R1) (Figure 1). 

### 2.3. PCR Analysis

To amplify each intron region, gradient PCR was used to optimize suitable conditions using an annealing temperature ranging between 50 and 60 °C. The PCR conditions were initial denaturation at 94 °C for 30 s, annealing at 50–60 °C for 30 s, extension at 72 °C for 1 min, and a final extension 72 °C for 5 min. In the case of low concentration (no/faint band), one microliter of the first PCR product was used as the DNA template for the second PCR using the same conditions as used with the first PCR. The PCR mixture contained 1× TaKaRa Ex PCR buffer, 0.2 mM dNTPs (each), 0.2 μM of each primer, and 1.0 U of TaKaRa Ex *Taq* polymerase (Takara Bio Inc., Shiga, Japan). Subsequently, the PCR product underwent electrophoresis in 1% agarose gel and was visualized with GelRed^TM^ Nucleic Acid Gel Stain (Biotium, Inc., Hayward, CA, USA). The PCR product (~1000 bp) was cut for gel purification using an E.Z.N.A.^®^ Gel Extraction kit (Omega bio-tek, Norcross, GA, USA) and was subsequently used for DNA sequencing and cloning.

### 2.4. DNA Sequencing and Molecular Cloning

The purified PCR products were cycle-sequenced via a commercial service provider (Eurofin, Tokyo, Japan). Sequencing results were visualized in a Sequence Scanner ver. 1.0 program to check for heterozygous patterns (Figure 2). Thereafter, heterozygous samples were detected from chromatograms, which showed the existence of indel nucleotide (Figure 2). Then, the purified PCR product of a particular heterozygous sample was cloned into a pGEM-T Easy Vector (Promega, Madison, WI, USA) according to manufacturer’s instructions. The recombinant plasmid was transformed and propagated in *Escherichia coli* JM109, and then four to ten inserted colonies were randomly picked and cultured in Luria-Bertani (LB) broth at 37 °C with continuous horizontal shaking for 12 h. Then, plasmid DNA was extracted using the FastGene^®^ Plasmid Mini kit (Nippon Genetics Co., Ltd., Tokyo, Japan) following manufacture’s protocol. The DNA from those plasmids was used as a template for sequencing in both directions using M13F and M13R as sequencing primers to detect the different haplotypes mixed in each particular heterozygous sample.

### 2.5. Data Analysis

Sequences were manually checked and edited using the BioEdit program version 7.2.6 [20]. Exon regions were blasted in NCBI for checking, and the real intron of TK was amplified. Thereafter, those exon regions were trimmed out before being used in intron sequence analysis. The intron sequences were multiple aligned using the ClustalW program version 2.0 [21] to compare and search for the variable sites, including microsatellite and insertion/deletion (indel) regions. Diversity indices and haplotype data were calculated and generated using the DnaSp program version 5 [22]. A neighbor joining (NJ) tree was constructed by using the MEGA program version 10 [23].

## 3. Results

### 3.1. Intron Amplification

Primer pairs were designed to amplify the seven intron regions, of these, five intron regions (TkD1Int2, TkD1Int5, TkD2Int2, TkD2Int4, and TkBridInt) were successfully amplified. From our experience, the suitable intron region for use as a genetic marker for genetic investigation in the trematodes should amplify a PCR product of 800–1200 bp in length. Thus, we selected only TkD1Int5, which provided a PCR product in length ~1000 bp. The other intron regions produced a PCR product either less than 800 bp or larger than 3000 bp (Appendix A). We tested the cross reaction of TkD1Int5 primers with the other *O. viverrini*-like eggs flukes, namely *H. taichui*, *H. pumilio*, and *S. fulcatus*, using the same PCR condition with *O. viverrini* and *C. sinensis* amplification. We found no cross-reaction. 

### 3.2. Nucleotide Sequence Analysis

After the exon sequences were trimmed, the length of TkD1Int5 used in this analysis ranged between 852 and 893 for *O. viverrini*, and was 871 bp for *C. sinensis*. All TkD1Int5 haplotype sequences of *O. viverrini* and *C. sinensis* examined in this study were deposited in GenBank under the accession numbers OR454257–OR454358 and OR440582–OR440599, respectively. Several insertion/deletion (indel) regions were observed in the TkD1Int5 region of *O. viverrini* (Appendix A). While only one nucleotide insertion was found in the TkD1Int5 region of *C. sinensis* from Vietnam (Appendix A). One repeated sequence/microsatellite “CTGTA” was embedded in the TkD1Int5 region of *O. viverrini*, and two large indel fragments, namely, fragments of nucleotide sites 356–372 and 423–450 were detected in *O. viverrini*. Based on variable sites observed in TkD1Int5 of 147 *O. viverrini* (185 sequences) and 53 *C. sinensis* sequences, 102 (Ov1–Ov102) and 18 (Cs1–Cs18) haplotypes were generated, respectively (Figure 3). 

Seven haplotypes of *O. viverrini* were shared between isolates, namely haplotypes Ov15, Ov16, Ov20, Ov31, Ov48, Ov52, and Ov74. The most common haplotype was Ov48, which was found in 34 sequences from various isolates (Figure 4). The TkD1Int5 sequence analysis of *C. sinensis* revealed three haplotypes, Cs1, Cs2, and Cs16, shared between different isolates/animal hosts. The isolates from humans depicted a high number of specific haplotypes, while no specific haplotypes were observed in *C. sinensis* collected from cats and dogs (Figure 4). 

### 3.3. Heterozygosity and Phylogenetic Tree Analysis

Several samples of *O. viverrini* contained indel and/or microsatellite regions which was the predominant heterozygous pattern observed in this study. The 37 *O. viverrini* samples exhibited heterozygous pattern(s) by direct DNA sequencing (Figure 2). All heterozygous samples were cloned and randomly picked inserted colonies for DNA sequencing. Of these, the 27 samples demonstrated the combination of at least two haplotypes within a particular heterozygous sample, while only one haplotype could be detected in the rest of the heterozygous samples. Interestingly, there were eight *O. viverrini* samples that contained three haplotypes within a particular heterozygous sample, namely CP8, KBp4, KLp5, KPv3, MD1, SK1, VV4, and VV6 (Figure 5). These haplotypes contained the indel regions and the repeated “CGCTA” region, which are the main evidence of heterozygosity (Figure 5).

We found a heterozygous pattern in *C. sinensis* with double peaks at one nucleotide site (Figure 6). This heterozygous pattern was found in the 15 *C. sinensis* recovered from humans, i.e., CSJ01–CSJ08, CSJ10–CSJ12, and CSJ15–CSJ18. The phylogenetic tree constructed by TkD1Int5 sequences revealed that 102 *O. viverrini* haplotypes clustered as a monophyletic group (Appendix A), as also did *C. sinensis* (Appendix A). 

## 4. Discussion

We successfully characterized a highly polymorphic intron region TkD1Int5, which can be used as a potential genetic marker for the investigation of the genetic variation and population genetics of *O. viverrini* sensu lato and *C. sinensis*. The TkD1Int5 region showed high polymorphism in *O. viverrini* but low variation in *C. sinensis*. This finding provided evidence that *C. sinensis* is more clonal than *O. viverrini*. In addition, the TkD1Int5 region of *O. viverrini* contained the repeated sequence and several indel regions, which caused the heterozygosity observed in this study. We unfortunately could not obtain *O. felineus*, another related species, to compare to *O. viverrini* and *C. sinensis*. The TkD1Int5 region could possibly be used for further genetic investigations of *O. felineus*. Interestingly, other intron and/or exon regions examined in this study could potentially be used to examine genetic differentiation between these liver flukes and other related species, especially co-endemic *O. viveriini*-like egg flukes. For instance, seven species of Southeast Asian *Paragonimus* could be differentiated by using variable nucleotide sites in the exon region of TK [16].

These liver flukes are diploid hermaphroditic trematodes. Therefore, reproduction can involve either self-fertilization or cross-fertilization [4]. Previous studies have commonly observed heterozygosity in *O. viverrini* by allozyme and microsatellite analyses, which indicates cross-fertilization as an optional mode of reproduction [7]. The results in our study found that 37 *O. viverrini* DNA samples (25.2%) depicted heterozygous patterns. Interestingly, seven *O. viverrini* DNA samples contained three different genotypes of TkD1Int5 within a particular sample. The samples of adult worms examined were from a hamster with an infected dose of 50–100 metacercariae, hence providing evidence that cross-breeding can occur in an infected hamster resulting in heterozygous fertilized eggs. In this case, DNA extraction was performed by using a whole adult worm. Therefore, DNA from fertilized eggs in their uterus can be extracted, which may have contaminated the total DNA sample. However, there is also the possibility that the triploid sample contained three haplotypes, as evidenced in *Paragonimus westermani* [16]. Only diploidy was observed in a previous chromosomal and karyotype analysis of *O. viverrini* [24], however, it was conducted on a small sample number of isolates of *O. viverrini*. A larger sample size from a wide variety of isolates and animal hosts needs to be investigated to provide a better understanding of ploidy.

Most recently, *O. viverrini* samples collected from Phang Khon District, Sakon Nakhon Province (SKp), have been previously classified as a cryptic genetic group by microsatellite, mitochondrial, and nuclear DNA markers [7]. We recently found that 10 *O. viverrini* samples from SKp could be classified into three haplotypes, i.e., Ov15, Ov31, and Ov33. Haplotype Ov15 was shared between the samples from several isolates, namely Lampang (LP), Khon Kaen (KPv and KBp), and SKp from Thailand; and Vientiane (VT), Khammouan (KM), Nam Ngum (NG), and Vang Viang (VV) from Lao PDR. Haplotype Ov31 was shared between LP and SKp, whereas Ov33 haplotypes were specifically detected only in SKp. However, in the phylogenetic tree, SKp was not clearly resolved as a genetically distinct group, as was seen when examining the TkD1Int5 region. Our observation raised an interesting question as to whether the SKp *O. viverrini* population consisted of two genetic groups.

Several closely related species of *O. viverrini* have recently been reported, i.e., *Opisthorchis lobatus* found in freshwater fish in Lao PDR, which may also cause zoonosis, but its role in humans is not known [25]. Similarly, the role of avian species is unknown; they include *O. cheelis*, *O. longissimus*, and *O. parageminus* which have been also reported from Southeast Asia [26,27]. Dao and colleagues [28] reported a sympatric distribution of duck and human genotypes of “*O. viverrini*”. The discovery of several species in the genus *Opisthorchis* in addition to the species complex of *O. viverrini* reflects complicated host and parasite interactions and potential co-evolution. Thus, comprehensive molecular systematics investigations of *O. viverrini* and its sibling species are needed using TK intron sequences or the other polymorphic DNA sequences as genetic markers.

As these liver flukes can infect multiple hosts [7], they are potentially zoonotic; however, whether this is the case remains controversial. For instance, a recent report found that *O. viverrini* from infected cats and humans, despite being genetically similar, separated into two distinct genetic groups, suggestive of host specificity of ‘human’ and ‘cat’ genotype [5]. Unfortunately, we did not have access to *O. viverrini* samples collected from other animal hosts to use in comparative analyses in our study. Further investigations should be conducted using TkD1Int5 for the geno typing of *O. viverrini* collected from various natural species of animal hosts. It has been shown that genetic variation of *C. sinensis* is not related to its infected hosts [6], but significant differences have been found between different geographical isolates from Russia and Vietnam [29]. We also found that *C. sinensis* from infected cats, dogs, rabbits, and humans were not genetically distinct, which was shown when examining by TkD1Int5. These results provide compelling evidence that *C. sinensis* performs a significant role in zoonotic transmission. However, we found a heterozygous pattern of *C. sinensis* recovered from humans. This finding suggested that the cryptic genotype(s) of *C. sinensis* in their natural hosts may exist; hence, a larger sample size from wide variety of natural species of hosts needs to be further investigated to determine whether this is the case.

## 5. Conclusions

This study reported genetic variation of *O. viverrini* and *C. sinensis* collected from various geographical isolates and animal hosts using the intron region TkD1Int5 as a genetic marker. A high variation of the TkD1Int5 sequence was observed in *O. viverrini*, while a low variation was observed in *C. sinensis*, reflecting that *C. sinensis* is more clonal than *O. viverrini*. Heterozygosity was observed in several *O. viverrini* samples, which indicates that cross-fertilization is an optional mode of reproduction, and that TkD1Int5 was suitable for population genetic analysis of *O. viverrini*. However, we found no genetic differences between *C. sinensis* recovered from different animal hosts. This finding demonstrated that various animal hosts play significant roles in maintaining the life cycle of *C. sinensis*. Hence, it is highly possible that *C. sinensis* is circulated between human and animal hosts. In the case of *O. viverrini*, we need further investigations of the TkD1Int5 sequence from samples collected from a range of different species of animal hosts across a wide geographical range to better understand its role in zoonotic diseases. If these mammalian hosts are involved in the transmission cycle of these liver fluke circulated between humans, then control approaches must also include zoonotic cycles in carnivore reservoir hosts, which are significant as they will maintain the prevalence of infection and incidence of disease in humans.

## Figures and Tables

**Figure 1 animals-13-03200-f001:**
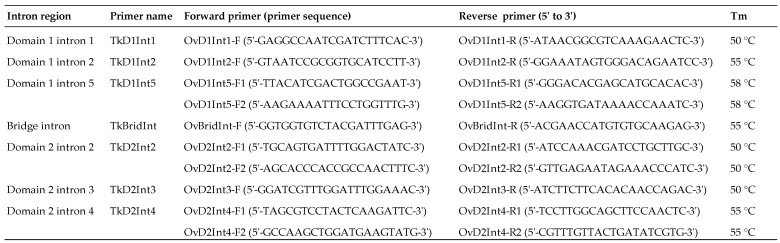
List of primers using for PCR amplification of seven intron regions of taurocyamine kinase gene. Tm is optimum annealing temperature for PCR of each primer pair.

**Figure 2 animals-13-03200-f002:**
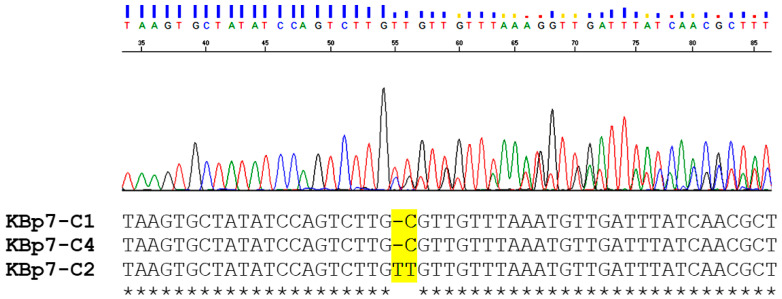
Heterozygous pattern of an *O. viverrini* sample KBp7 represents an insertion/deletion (indel) region as yellow highlighted. After the indel, the chromatogram of nucleotide sequence is unreadable. Three clones and plasmid sequencing revealed that different haplotypes of TkD1Int5 sequence were mixed in this sample. Identical nucleotide indicated by “*”. Different line colors represent different nucleotide base, red: base T; blue: base C; greed: base A; black: base G.

**Figure 3 animals-13-03200-f003:**
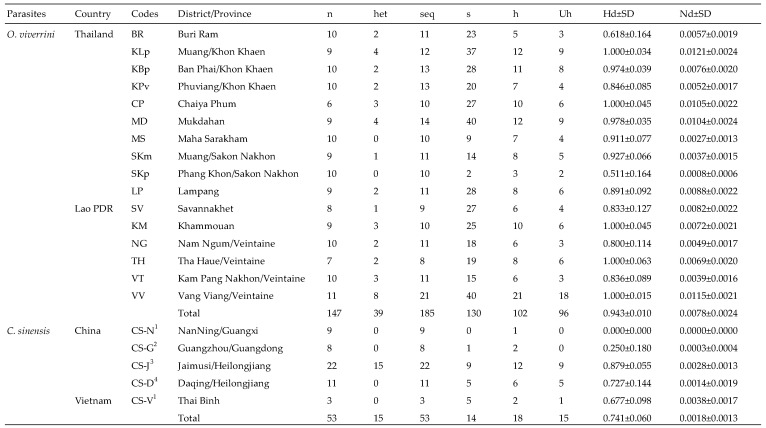
Diversity indices of the sequences of domain 1 intron 5 of taurocyamine kinase gene in the *O. viverrini* and *C. sinensis* populations from various isolates and animal hosts. Isolates from naturally infected cats ^1^, dogs ^2^, humans ^3^, and experimental rabbits ^4^, whereas all *O. viverrini* samples were from experimental hamsters. n: Number of DNA sample; het: number of heterozygous samples; seq: number of sequences generated in a particular isolate; s: segregations site; h: number of haplotypes; Uh: number of unique haplotypes; Hd: haplotype diversity; Nd: nucleotide diversity; SD: standard deviation.

**Figure 4 animals-13-03200-f004:**
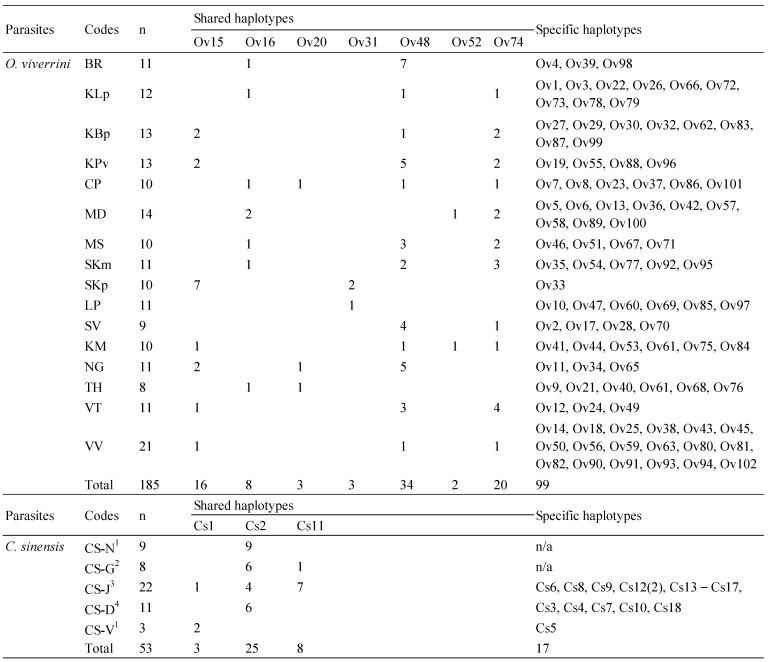
Frequency of shared and specific haplotypes of *O. viverrini* and *C. sinensis* populations examined by domain 1 intron 5 of taurocyamine kinase gene sequence. Isolates from naturally infected cats ^1^, dogs ^2^, humans ^3^, and experimental rabbits ^4^, whereas all *O. viverrini* samples were from experimental hamster. n: Number of sequences.

**Figure 5 animals-13-03200-f005:**
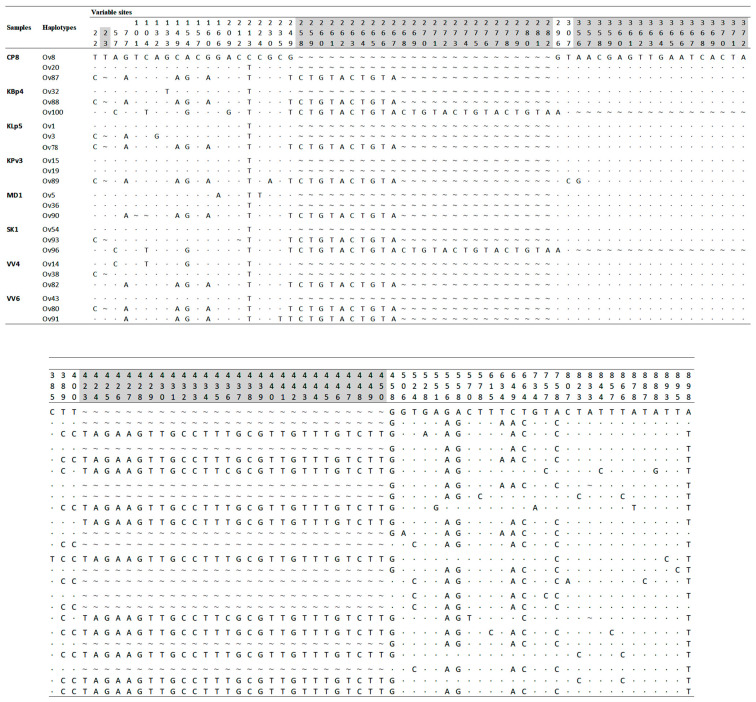
Variable nucleotide positions of TkD1Int5 of eight *O. viverrini* samples contained three haplotypes in a particular sample. Identical nucleotide indicated by “·”; insertion/deletion (indel) site indicated by “~”; indel site leading to heterozygosity observation indicated by gray shedding.

**Figure 6 animals-13-03200-f006:**
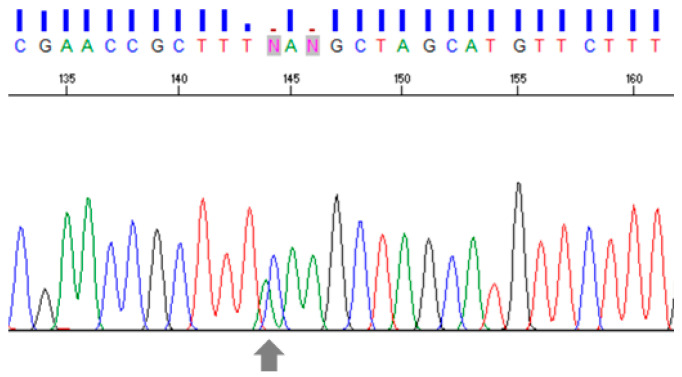
Heterozygous pattern of *C. sinensis* represents double peaks (A/C) at a particular nucleotide site as pointed by an arrow. Different line colors represent different nucleotide base, red: base T; blue: base C; greed: base A; black: base G.

## Data Availability

All data are available upon request.

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
