# Peer review of "Intron Regions as Genetic Markers for Population Genetic Investigations of Opisthorchis viverrini sensu lato and Clonorchis sinensis"

_animals, 2023, doi:10.3390/ani13203200_

Round 1

Reviewer 1 Report

Table 2: Table headings are not properly defined. Please mention what Hd and Nd represent.

It is unclear regarding the source of the genes used in Table 2 and Table 3. Please elaborate on the acquisition of these genes from the sources and the sampling criteria used. It appears the sample pool is rather restrictive.

What is the rationale to why O vivierrini was only isolated from experimental hamsters while C. sinensis was isolated from multiple human and animal hosts?

Is there sufficient evidence in this manuscript to demonstrate that C. sinensis plays an important role in zoonotic transmission? Direct transmission from vertebrate hosts is unlikely. Describing the geographical regions and whether there is a shared water source that facilitates reservoir host growth would be a better approach. The conclusion is rather loose.

Author Response

Reviewer 1#

Table 2: Table headings are not properly defined. Please mention what Hd and Nd represent.

Reply: Table 2 heading has been modified. Hd, Nd and the others have been clarified in footnote.

It is unclear regarding the source of the genes used in Table 2 and Table 3. Please elaborate on the acquisition of these genes from the sources and the sampling criteria used. It appears the sample pool is rather restrictive.

Reply: Source of the gene used in Table 2 and 3 are mentioned in the headings.

What is the rationale to why O vivierrini was only isolated from experimental hamsters while C. sinensis was isolated from multiple human and animal hosts?

Reply: We used DNA samples leftover from the previous studies. Thus, we cannot specify the source of sample isolates in our study. However, we have refered to this point in the conclusion that O. viverrini isolated from various species of animal hosts should be investigated in the future.

Is there sufficient evidence in this manuscript to demonstrate that C. sinensis plays an important role in zoonotic transmission? Direct transmission from vertebrate hosts is unlikely. Describing the geographical regions and whether there is a shared water source that facilitates reservoir host growth would be a better approach. The conclusion is rather loose.

Reply: We have modified the conclusion related to this point as “various animal hosts play significant roles in maintaining life cycle of C. sinensis”.

Reviewer 2 Report

Dear Authors

Your study, “Characterization of intron regions and their potential as a genetic marker for population genetic investigations of zoonotic liver flukes Opisthorchis viverini sensu lato and Clonorchis sinensis from various geographic isolates and animal hosts”, is relevant because it identifies a highly polymorphic genetic marker, which could be used on future studies of  regarding population genetics of trematode species with a wide geographic distribution and variety of animal hosts, classified as agents responsible for cholangiocarcinoma (CCA), more specifically in Asian countries.

My comments are the following:

1.      The Introduction provides an information on the subject under study, including relevant references.

2.      Material and Methods

2.1. Primer design

I suggest accession number JX435779 (line 134) to be replaced by the author of the article. The search will be much easier.

Xiao JY, Lee JY, Tokuhiro S, Nagataki M, Jarilla BR, Nomura H, Kim TI, Hong SJ, Agatsuma T. Molecular cloning and characterization of taurocyamine kinase from Clonorchis sinensis: a candidate chemotherapeutic target. PLoS Negl Trop Dis. 2013 Nov 21;7(11): e2548. doi: 10.1371/journal.pntd.0002548. PMID: 24278491; PMCID: PMC3836730.

In Line 148 … “annealing temperature ranging between 45-60ºC. Below, in Line 149, the annealing temperature ranging between 40-60ªC. “. Is it correct? In Table 1 the optimum annealing temperature for PCR primer pair ranges between 50-58º C. It's a little confusing.

3.      3.      Results

3.1. Intron amplification

Line 194 – Figure S1 should contain a legend to better understand the results.

3.2. Nucleotide sequence analysis

In Lines 209-211, I cannot found in Table 3 the number of sequences and isolates that are described in the text.  I suggest that this point should be better clarified.

 I suggest checking the language and spelling, considering that English is not my native language.

Author Response

Reviewer 2#

Your study, “Characterization of intron regions and their potential as a genetic marker for population genetic investigations of zoonotic liver flukes Opisthorchis viverini sensu lato and Clonorchis sinensis from various geographic isolates and animal hosts”, is relevant because it identifies a highly polymorphic genetic marker, which could be used on future studies of regarding population genetics of trematode species with a wide geographic distribution and variety of animal hosts, classified as agents responsible for cholangiocarcinoma (CCA), more specifically in Asian countries.

My comments are the following:

  1. The Introductionprovides an information on the subject under study, including relevant references.

Reply: Thank you.

  1. Material and Methods

2.1. Primer design

I suggest accession number JX435779 (line 134) to be replaced by the author of the article. The search will be much easier.

Xiao JY, Lee JY, Tokuhiro S, Nagataki M, Jarilla BR, Nomura H, Kim TI, Hong SJ, Agatsuma T. Molecular cloning and characterization of taurocyamine kinase from Clonorchis sinensis: a candidate chemotherapeutic target. PLoS Negl Trop Dis. 2013 Nov 21;7(11): e2548. doi: 10.1371/journal.pntd.0002548. PMID: 24278491; PMCID: PMC3836730.

Reply: We added a reference here as suggested.

In Line 148 … “annealing temperature ranging between 45-60ºC. Below, in Line 149, the annealing temperature ranging between 40-60ªC. “. Is it correct? In Table 1 the optimum annealing temperature for PCR primer pair ranges between 50-58º C. It's a little confusing.

Reply: Annealing temperature optimization was corrected to 50-60ºC.

  1.  Results

3.1. Intron amplification

Line 194 – Figure S1 should contain a legend to better understand the results.

Reply: Legend of Figure S1 has been modified.

3.2. Nucleotide sequence analysis

In Lines 209-211, I cannot found in Table 3 the number of sequences and isolates that are described in the text.  I suggest that this point should be better clarified.

Reply: Number of sequences and isolates in the text have been corrected.

 I suggest checking the language and spelling, considering that English is not my native language.

Reply: English has been improved by two native speakers, Prof. Ross H Andrews and Dr. Adrian Plants.

Reviewer 3 Report

Thank you for shedding light on population genetics of liver flukes with extensive data collection and analysis. Please consider shortening the title while including the important details related to the article. A suggestion after reading the manuscript would be "Intron regions as genetic marker for population genetic investigations of Opisthorchis viverrini sensu lato and Clonorchis sinensis".

In this manuscript, the author has characterized seven intron regions of taurocyamine kinase (TK) gene as molecular markers to perform population genetic studies of two zoonotic liver flukes: Opisthorchis viverrini sensu lato and Clonorchis sinensis. The rationale of this study is mentioned in the introduction section, where the author has referred to previous publications studying intron regions of TK in other medically important trematodes. The author has however not mentioned the advantages (if any) of using TK intron regions of Opisthorchis viverrini sensu lato and Clonorchis sinensis over other conventional molecular markers (mitochondrial or ribosomal) widely used in other organisms.

Similarly, the author has not mentioned or used a positive control and an internal control primer set to validate using the intronic regions as a novel molecular marker and to depict equal loading in an agarose gel, respectively. The author is advised to either provide such a figure to supplement the study or provide a strong justification of the absence of this data. It will be informative to learn from the author about polymorphisms in the exon regions of the TK gene and if liver flukes are known to translate splice variants of TK. Additionally, the author can discuss in the manuscript if liver fluke TK protein possess any immune pressure leading to known synonymous or non-synonymous mutations in the protein coding region? If yes, has there been a correlative study on presence of such SNPs in protein coding regions with the presence of polymorphic regions in the introns studied here by the authors?

Two sentences used in the conclusion section either need more description to support how these conclusions are drawn from this study, or need to be adjusted to avoid chances of overstating from the results. The sentences are in the line 337, where cross fertilization is suggested to be 'preferable' by the author. Another is mentioned in line 340, where C. sinensis is suggested to be 'important for zoonotic transmission' due to no genetic differences observed in the study. Is it possible that more proteins (exon/intron regions) need to be explored to understand population genetic differences in C. sinensis?

A minor point for the author would be to consider providing a list of abbreviations for the ones used in the Table 2, to help the readers with the jargons used in the table.

Over all, the manuscript is written in scientifically accurate language. However, there are multiple grammatical and typographical errors that can be confusing for interpretation. Some of the sentences that can be improved are in line 27, 41, 82, 96, 184, 196, 282, 314, 316, 317, and 318. The author is advised to use text editing for further steps leading to publication, especially in these sentences.

Author Response

Reviewer 3#

Thank you for shedding light on population genetics of liver flukes with extensive data collection and analysis. Please consider shortening the title while including the important details related to the article. A suggestion after reading the manuscript would be "Intron regions as genetic marker for population genetic investigations of Opisthorchis viverrini sensu lato and Clonorchis sinensis".

Reply: We have modified the title a suggested.

In this manuscript, the author has characterized seven intron regions of taurocyamine kinase (TK) gene as molecular markers to perform population genetic studies of two zoonotic liver flukes: Opisthorchis viverrini sensu lato and Clonorchis sinensis. The rationale of this study is mentioned in the introduction section, where the author has referred to previous publications studying intron regions of TK in other medically important trematodes. The author has however not mentioned the advantages (if any) of using TK intron regions of Opisthorchis viverrini sensu lato and Clonorchis sinensis over other conventional molecular markers (mitochondrial or ribosomal) widely used in other organisms.

Reply: The advantages of using an intron region of Tk has been additionally mentioned in introduction section in line 111 – 119.

Similarly, the author has not mentioned or used a positive control and an internal control primer set to validate using the intronic regions as a novel molecular marker and to depict equal loading in an agarose gel, respectively.

Reply: We have checked that our PCR product is a true intronic region of TK gene by blasting a flanking region of exon sequence (~50 bp), and it was 100% similar to TK gene in NCBI.

The author is advised to either provide such a figure to supplement the study or provide a strong justification of the absence of this data.

Reply: We have provided a figure of PCR products of five intron regions that can be amplified. Only the PCR product size ranging between 800 – 1200 bp is the most suitable as genetic marker for further investigations. Therefore,  only the TkD1Int5 was selected in this study.

It will be informative to learn from the author about polymorphisms in the exon regions of the TK gene and if liver flukes are known to translate splice variants of TK. Additionally, the author can discuss in the manuscript if liver fluke TK protein possess any immune pressure leading to known synonymous or non-synonymous mutations in the protein coding region? If yes, has there been a correlative study on presence of such SNPs in protein coding regions with the presence of polymorphic regions in the introns studied here by the authors?

Reply: We did not compare the exon sequence in this study. However, as the exon region can be potentially used to differentiate species of Southeast Asian lung fluke genus Paragonimus (Tantrawatpan et al., 2021), the exon sequence may be used in further investigations to determine if it can discriminate O. viverrini and the other O. viverrini-like eggs.

Tantrawatpan, C., Tapdara, S., Agatsuma, T., Sanpool, O., Intapan, P.M., Maleewong, W., Saijuntha W. Genetic differentiation of Southeast Asian Paragonimus Braun 1899 (Digenea: Paragonimidae) and genetic variation in the Paragonimus heterotremus complex examined by nuclear DNA sequences. Infect. Genet. Evol. 2021. 90, 104761.

Two sentences used in the conclusion section either need more description to support how these conclusions are drawn from this study, or need to be adjusted to avoid chances of overstating from the results. The sentences are in the line 337, where cross fertilization is suggested to be 'preferable' by the author.

Reply: We use the word “optional” instead of “preferable” to avoid overstating result.

Another is mentioned in line 340, where C. sinensis is suggested to be 'important for zoonotic transmission' due to no genetic differences observed in the study. Is it possible that more proteins (exon/intron regions) need to be explored to understand population genetic differences in C. sinensis?

Reply: We have modified a conclusion related to this point as “various animal hosts play significant roles in maintaining the life cycle of C. sinensis”.

A minor point for the author would be to consider providing a list of abbreviations for the ones used in the Table 2, to help the readers with the jargons used in the table.

Reply: All abbreviations in Table 2 have been clarified in the footnote.

Over all, the manuscript is written in scientifically accurate language. However, there are multiple grammatical and typographical errors that can be confusing for interpretation. Some of the sentences that can be improved are in line 27, 41, 82, 96, 184, 196, 282, 314, 316, 317, and 318. The author is advised to use text editing for further steps leading to publication, especially in these sentences.

Reply: Those sentences are corrected by two native speakers, namely Prof. Ross Andrews and Dr. Adrian Plants.

Round 2

Reviewer 1 Report

The authors sufficiently justified their observations and made the necessary corrections. 

Author Response

Thank you for accepting our manuscript.

Reviewer 3 Report

Author has provided changes asked as asked for in the comments

Author Response

Thank you for accepting our manuscript.